# Agar Biopolymer Films for Biodegradable Packaging: A Reference Dataset for Exploring the Limits of Mechanical Performance

**DOI:** 10.3390/ma15113954

**Published:** 2022-06-01

**Authors:** Valentina Hernández, Davor Ibarra, Johan F. Triana, Bastian Martínez-Soto, Matías Faúndez, Diego A. Vasco, Leonardo Gordillo, Felipe Herrera, Claudio García-Herrera, Alysia Garmulewicz

**Affiliations:** 1Department of Management, Faculty of Management and Economics, University of Santiago of Chile (USACH), Avenida Libertador Bernardo O’Higgins 3363, Estación Central, Santiago 9170022, Chile; valentina.hernandezm@usach.cl; 2Department of Mechanical Engineering, University of Santiago of Chile (USACH), Avenida Libertador Bernardo O’Higgins 3363, Santiago 9170022, Chile; davor.ibarra@usach.cl (D.I.); matias.faundez@usach.cl (M.F.); diego.vascoc@usach.cl (D.A.V.); claudio.garcia@usach.cl (C.G.-H.); 3Department of Physics, University of Santiago of Chile (USACH), Avenida Victor Jara 3493, Santiago 9170124, Chile; johan.triana@usach.cl (J.F.T.); leonardo.gordillo@usach.cl (L.G.); felipe.herrera.u@usach.cl (F.H.); 4Department of Mathematics and Computer Science, University of Santiago of Chile (USACH), Las Sophoras 173, Santiago 9170124, Chile; bastian.martinez.s@usach.cl; 5ANID-Millennium Institute for Research in Optics, Concepción 4030000, Chile; 6CABDyN Complexity Centre, University of Oxford, Oxford OX1 2JD, UK

**Keywords:** bioplastic, agar, biodegradable packaging, seaweed, mechanical properties, machine learning

## Abstract

This article focuses on agar biopolymer films that offer promise for developing biodegradable packaging, an important solution for reducing plastics pollution. At present there is a lack of data on the mechanical performance of agar biopolymer films using a simple plasticizer. This study takes a Design of Experiments approach to analyze how agar-glycerin biopolymer films perform across a range of ingredients concentrations in terms of their strength, elasticity, and ductility. Our results demonstrate that by systematically varying the quantity of agar and glycerin, tensile properties can be achieved that are comparable to agar-based materials with more complex formulations. Not only does our study significantly broaden the amount of data available on the range of mechanical performance that can be achieved with simple agar biopolymer films, but the data can also be used to guide further optimization efforts that start with a basic formulation that performs well on certain property dimensions. We also find that select formulations have similar tensile properties to thermoplastic starch (TPS), acrylonitrile butadiene styrene (ABS), and polypropylene (PP), indicating potential suitability for select packaging applications. We use our experimental dataset to train a neural network regression model that predicts the Young’s modulus, ultimate tensile strength, and elongation at break of agar biopolymer films given their composition. Our findings support the development of further data-driven design and fabrication workflows.

## 1. Introduction

Plastics have become a major contaminant in ecological systems due to lack of biodegradability, low recycling rates, and poor waste management practices. 80–85% of all marine pollution is attributed to plastic [1], and single-use packaging, including lightweight plastic bags and plastic films, making up half of all marine plastic pollution [2]. Single-use plastics often break down into microplastic particles; these are of growing concern as a worldwide contaminant in ocean and freshwater ecosystems, where they can bioaccumulate and carry toxins [3]. In 2018, 359 million tons of plastic were produced, the vast majority from non-renewable feedstock [4]. This problem is projected to become more acute, as plastics are predicted to account for 20% of oil consumption by 2050 [5].

Bioplastics that are biodegradable and biobased offer promising alternatives, specifically for single-use, flexible packaging applications that have high rates of leakage [2]. However, in the development of bioplastic alternatives, it is imperative to avoid the use of feedstock that compete for arable land for food and feed and incur land-use emissions (i.e., corn, oil, and sugar crops) [6]. Seaweeds have been recognized as sustainable feedstock for producing bioplastics given that they do not require freshwater, arable land, or fertilizer to grow, they uptake excess nutrients from seawater, and act as carbon sinks, thus having a mitigating effect on climate change [7]. Studies have shown the viability of producing a range of bioplastics and biocomposites from red (Rhodophyta) and brown (Phaeophyta) seaweed through the extraction of agar, carrageenan, alginate, long-chain hydrophilic polysaccharides that form gels when dispersed in water [8]. Additional components extracted from seaweed include the intracellular components of starch, cellulose, and algal proteins [9]. Applications include packaging such as coatings and thin films [9,10,11,12,13]. Moreover, seaweed bioplastics have been found to biodegrade in soil in only four to six weeks and do leave microplastic residue [14]. The promise of seaweed-based bioplastics for thin film, single-use packaging motivates the present study.

Despite studies investigating agar in combination with other polymers and additives [15], there has been a lack of attention paid to understanding the mechanical performance that agar alone can achieve in combination with simple plasticizers, with experiments limited to a small set of formulations [16,17,18,19]. While it is asserted that simple agar films are relatively brittle, studies often include only one or two reference formulations of agar and a plasticizer such as glycerol or sorbitol, while most of the focus is on the performance of more complex formulations [20,21]. This lack of data limits a more comprehensive understanding of the design space for agar-based biopolymer films for packaging applications.

First, we introduce our methodology for developing an original study of the mechanical properties of agar-glycerin biopolymer films. We take a Design of Experiments approach to analyze how agar-glycerin biopolymer films perform across a range of ingredients concentrations in terms of their strength, elasticity, and ductility, properties central to single-use packaging applications. Second, we compare our results to previous studies of more complex agar biopolymer films that include other additives, using the review provided by Mostafavi and Zaeim [15] as a comparative dataset to situate our study. This review includes studies of agar blended with a range of other biopolymers, organic and inorganic nanoparticles, and oils. To better utilize it as a comparative dataset, original data ranges for mechanical properties were extracted from each paper in the review, expanding the summary information initially reported [15] (see Appendix B). We find that by systematically varying the quantity of agar and glycerin, certain tensile properties can be achieved for a simple agar biopolymer film that are comparable to those with more complex formulations in terms of additives and processing. Third, we compare our data to commercial petroleum plastics presently used for packaging films, finding that select formulations have tensile properties that are comparable to thermoplastic starch (TPS), acrylonitrile butadiene styrene (ABS), and polypropylene (PP). Fourth, we use our experimental dataset to train a neural network regression model that predicts the Young’s modulus, ultimate tensile strength, and elongation at break of agar biopolymer films given their composition. Finally, we discuss our results and the need for data-driven design and fabrication workflows for the development of biodegradable plastics packaging.

## 2. Materials and Methods

### 2.1. Experimental Design and Data Analysis

We use a factorial experimental design, or Design of Experiments (DoE) approach that allows one to explore, map, and model the behavior of a material system within an experimental range [22]. DoE is commonly used in polymer research [23], and factorial designs are widely used in experiments in which several factors have a combined effect on a variable of interest [24]. More specifically, we use a 2^k^ Factorial Design methodology to evaluate the effect of the percentage of agar and glycerin on the response variables of Young’s modulus, ultimate tensile strength, and elongation at break of the biopolymer films, important properties for packaging applications. We group the data in three segments and identify a high and a low point for each data range. For each data range we carry out an analysis of variances and perform a nonlinear regression of the form
Y = β_0_ + β_1_x_1_+ β_2_x_2_ + β_3_x_1_x_2_ + *E*,(1)
where Y represents the response variable studied (elongation at break, ultimate tensile strength, and Young’s modulus), β_0_ denotes the model intercept, β_1_, β_2_, and β_3_ denote the coefficients corresponding to percentages of agar, glycerin, and the interaction between them respectively, and *E* denotes the error term. We also use our experimental dataset to train a neural network regression model that predicts the mechanical properties of agar-based biofilms given their composition. More details about these data analysis techniques are given in Section 2.6.

### 2.2. Materials

Agar consists of two polysaccharides, agarose and agaropectin, and forms a structural component of the cell walls of red seaweed. Agar can be extracted by washing the seaweed, heating the seaweed in water to dissolve the agar, filtering to remove residuals, cooling the filtrate to form a gel, and then removing the water from the gel via pressure or a freeze-thaw process [11,25].

The ranges of ingredients concentrations used to make the biopolymer films in this study are as follows: agar (A), 0.33% to 8% [mL/mL of water], and glycerin (G) 0.21% to 10.7% [mL/mL of water]. Demineralized water (W) makes up the remaining proportion across the range of formulations. Glycerin and water act as plasticizers and help give flexibility to the final agar-based films [26,27]. Agar was sourced from Cherry Chile S.A. Per 100 g, the agar contains 76.1% carbohydrates, 0.1% fat, 2.3% protein, 20.1% water, 800 mg sodium, 400 mg calcium, 5 mg iron, 8 mg phosphorus, and 1.4 mg iodine. USP grade vegetable glycerin (approved for pharmaceutical use) was used, with 99.7% purity.

### 2.3. Preparation of Agar Biopolymer Films

Agar was added to a mixture of glycerin and water and heated to a temperature of 90 °C using a magnetic stirrer and heater. The combined solution was then poured into silicone molds and dried over a period of approximately three days. To form biopolymer films, the solution undergoes condensation polymerization. Gelation occurs upon cooling the mixture, where the components polymerize via cross-linking. At the gel point, where the solution undergoes a rapid change in viscosity, moving from liquid to solid, the average temperature was 21.68 °C (min 19.9 °C, max. 23.2 °C) and the relative humidity was 53.4% (min 45.1%, max 67.3%). These measures were taken with a TESTO digital thermohygrometer with a resolution of ±3% RH and ±0.5 °C. The content of the resulting biopolymer film is a combination of the precursors, agar, glycerin, and water, with a loss of water content.

### 2.4. Experimental Design

A Design of Experiments (DoE) approach was taken using a full factorial mixture design using the methodology 2*k* where *k* is the number of factors considered at two levels. This design allows one to identify which variables in the process have the greatest influence on the behavior of the material. Three factors were specified (agar, glycerin, and water) at two levels (“high” +1, and “low” −1). Table 1 shows the input data for the DoE.

Formulations with less than 3% agar and 0.313% glycerin were not included in the data. At these ingredient concentrations, the manufacturing process became infeasible due to accelerated solidification, preventing the pouring of a homogeneous mixture into the molds. High amounts of agar significantly modified the physical characteristics of the material, resulting in a thick rubber-like material that was not suitable for tensile testing.

### 2.5. Characterization of Mechanical Properties

Tensile tests were performed on an INSTRON universal test machine with a load cell of up to 500 N. The apparatus was fitted with a clamping system ideal for thin-film materials, consisting of two plates with adherent surfaces controlled by a pneumatic system that allows for the maintenance of constant pressure (more than 50 psi) by the clamps.

The sample specimens were cut from sheets of biopolymer film using a die with dimensions shown in the accompanying dataset [28]. While the dimensions of the die remained constant, film thickness varied by formulation. This was measured with a MITUTOYO digital micrometer with a resolution of ±0.001 mm. A dog bone shaped specimen was used as it better allowed stress to concentrate in the area with the least cross-sectional area, favoring more accurate calculation of mechanical properties. Otherwise, the test procedure was carried out according to ASTM D882 standard for plastic film. The force measured by the load cell was transformed into units of Megapascals (MPa), divided by the average cross section of the specimens in m^2^. The dimensionless axial deformation was calculated as the percentage of elongation with respect to the distance between the jaws, or the initial length of the specimen. For each concentration, an average of five test specimens were obtained from the same sheet for tensile testing. The dataset was scaled by simple interpolation to obtain corresponding average values. Table 2 provides an overview of the formulations characterized.

### 2.6. Neural Network Regression

Machine learning can be expected to be useful for predicting novel formulations with target material properties, based on a model trained on empirical datasets [3]. Once a neural network is trained and validated with a suitable experimental dataset, it can be used to predict the properties of unknown material formulations. These can then be synthesized and measured to either validate the neural network model or suggest model improvements. This approach has recently been used for mapping the multi-dimensional structure-property parameter space of common polymers [29].

We train an artificial neural network to predict the mechanical properties of agar biopolymer films using ingredient concentrations as the input. The neural network regression model is trained with the data generated from the Design of Experiments (DoE) approach described in Section 2.4, with an additional 17 experiments added for increased model effectiveness. The dataset contains 160 records with agar (A), glycerin (G), and water (W) concentrations as input features, and Young’s modulus (*E*), ultimate tensile strength (UTS), and elongation at break (ε) as output features. The input data corresponds to 32 distinct formulations of agar, glycerin, and water, with an average of five samples per formulation. The dataset used for training the regression model is publicly available at https://data.mendeley.com/datasets/nyz4y58jbt/2 (accessed on 19 May 2022).

The experimental dataset was converted to a Pandas Data Frame and the neural network was implemented with the Keras library. The input parameters were scaled, and five-fold cross-validation with 500 training epochs was implemented with 80% of the data used for training and 20% used for validation. The architecture consists of a densely connected network with an input layer of three neurons (A, G, W), followed by three hidden layers with 32 neurons each, and an output layer with three neurons (*E*, UTS, ε). A ReLu activation function was employed with Adam hyperparameter optimization. Mean squared error (MSE) was used for the loss function.

## 3. Results

First, we review the results from the tensile tests performed, noting the ranges of values obtained for Young’s modulus, ultimate tensile strength, and elongation at break. We then analyze the results of the statistical models, revealing how various ingredient factors are influenced at different concentrations in terms of resulting mechanical properties. This is followed by the results of the neural network model. We then explore how the mechanical properties of our materials compare to other agar-based biopolymer films in the scientific literature, in addition to the mechanical properties of commercial polymers.

### 3.1. Mechanical Property Ranges

A uniaxial tensile test was used to evaluate the stress and deformation of each material specimen. An average of five specimens were tested for each biopolymer film formulation, and the resulting data were normalized and interpolated to obtain the corresponding average stress and strain. The graph in Figure 1 shows a typical example of an average curve obtained in the tensile tests; this sample corresponds to a concentration of 2% agar and 1.25% glycerin, one from which higher measures of elongation at break and ultimate tensile strength were obtained. The curve shows rather ductile behavior, with low yield strength and resilience, and a highly plastic zone. Using linear regression, the linear part of the curves is used to calculate Young’s modulus, with a correlation factor of not less than 99.9%.

To select material formulations that meet the property requirements of specific applications, one must consider various trade-offs between material properties. Table 3 shows the relationships between measured mechanical properties by noting the minimum and maximum values for each variable tested (elongation at break, UTS and Young’s modulus) across the range of formulations fabricated.

When each material property is at its maximum or minimum value, one can view how this corresponds to the levels of the other properties, as well as the composition of the formulation. One notes that the formulation with the maximum value of Young’s modulus also has the maximum value of ultimate tensile strength, indicating a strong, stiff material. However, the value of elongation at break for this formulation is 14.68%, a comparatively low value. The minimum values for ultimate tensile strength and Young’s modulus also correspond to the same formulation. For elongation at break, higher percentages of plasticizer (glycerin) and agar increase material ductility. However, the material’s performance with regards to ultimate tensile strength and Young’s modulus are reduced, in line with other studies of polymer behavior [30].

With these property ranges, one can identify formulations that meet specific performance requirements. To identify formulations that may be relevant for specific applications, it is necessary to understand the mechanical stresses to which the material will be subjected to in the expected application. For instance, polymeric packaging films are often selected for high ductility and relatively high resistance to breakage [31].

### 3.2. Statistical Results of Experimental Design

To identify formulations that meet the functional requirements of an intended application, it is important to know how the material system responds to differing ingredients concentrations. Across a range of possible mixtures, the ingredients of a bioplastic may interact in ways that produce unexpected performance outcomes, measured in terms of material strength, elasticity, or ductility. By demonstrating the range of properties that are obtainable with these agar biopolymer films, our aim is to provide the necessary information for researchers and practitioners who seek to optimize specific formulations for product applications.

An analysis of variance was performed for the effect of ingredient concentrations on the mechanical properties of biopolymer films. The data were demarcated in three data ranges, according to low, medium, and high ingredients concentrations. Models for predicting Young’s modulus, ultimate tensile strength, and elongation at break were then run for each of the data segments. For the first data range, with low concentrations of both agar and glycerin, we note that a *p*-value value of less than 0.05 is obtained for each variable (% agar and % glycerin) and for the interaction between them. This indicates that both variables and their interaction have statistically significant effects on each of the mechanical properties. Model fit, or R^2^, ranged between 82.3 and 98.6%, indicating the percentage variation in mechanical properties that can be explained by the independent variables chosen. Variation in the percentage of agar was shown to be the most influential factor in the models for each mechanical property studied.

For the second data range, with medium concentrations of both agar and glycerin, the *p*-values for the percentage of agar and the combination of agar and glycerin in the model for elongation at break are greater than 0.05, indicating a lack of statistical significance. The value of R^2^ is low at 38.5%. For the Young’s modulus and UTS models, the *p*-values indicate that variation in the percentage of agar, percentage of glycerin and the relationship between them are statistically significant at a *p*-value of <0.05. It is also interesting to note that there is less variability when it comes to values of elongation at break compared to the variability obtained when measuring Young’s modulus and UTS.

For the third data range, with high concentrations of both agar and glycerin, the percentage of glycerin is found not to be statistically significant at a *p*-value level of <0.05 in predicting Young’s modulus, while the percentage of agar and the combination of agar and glycerin are. The value of R^2^ is 61%. For the models of elongation at break and ultimate tensile strength, all the factors and the relationship between them are statistically significant at a *p*-value of <0.05. The R^2^ values are 93.78% and 96.46% respectively. Overall, we can note that 7 of the 9 analyses performed gave a high level of model fit, indicating that most of the variation in mechanical properties is explained by the independent factors of ingredient percentages. Unexplained variation can be attributed to process and measurement variation. The complete regression tables and associated figures are provided in Appendix A.

As part of the analysis, we produce contour plots to observe the behavior of the response variables to different ranges of ingredients concentrations. As illustrated in Figure 2 for the response variable of elongation at break, at low and high ranges of ingredients concentrations a similar pattern is observed: higher values are found at lower ranges of agar and higher concentrations of glycerin.

At medium concentrations of both components, the value is constant. By contrast, the plots illustrating the response of ultimate tensile strength to low, medium, and high ingredients ranges show an uneven pattern. In the low range, the highest values of ultimate tensile strength correspond to a percentage of agar at 0.8–1%, and glycerin below 0.5%. In the medium range, high values of ultimate tensile strength are found in two regions: where agar is at 2% and glycerin at 1.3%, and where agar is at 3% and glycerin at 1.8%. In the high range, ultimate tensile strength is maximized when agar is at 8% and glycerin at 10%. This pattern reveals the importance of exploring a wide range of ingredients concentrations when assessing how certain ingredients influence a material property of interest. While analysis of the low range of ingredients concentrations appears to show that ultimate tensile strength increases when the percentage of agar increases, combining the plots reveals various optimal ranges. One notes that the optimal ranges for ultimate tensile strength across the three data segments correspond to an agar-glycerin ratio of approximately 60%, indicating that the relationship between these variables is critical for the pursuit of optimizing this mechanical property. The behavior of Young’s modulus is found to be quite like that of ultimate tensile strength.

### 3.3. Comparison to Other Agar-Based Biopolymer Films

The table in Appendix B details the mechanical properties of agar-based biopolymer films for food packaging applications reviewed by Mostafavi and Zaeim [15]. Here, we extract the full range of values available in each paper reviewed for better comparison with the results of this study. This comparison to our original dataset is illustrated in Figure 3, with original data demarcated in black and labelled ‘Agar’, with error bars signifying the range of values around property measurements.

Comparison of the dataset developed in this study with other agar-based biopolymer films reviewed by Mostafavi and Zaeim [15] offers mixed results. Figure 3a illustrates how adding other ingredients can clearly improve the Young’s modulus of agar-based biopolymer films; for example, those with added ingredients such as nanoclay and nanocellulose clearly perform better than the more basic formulations in this study.

However, Figure 3b shows that our data cover a significant range of ultimate tensile strength and elongation at break values compared to more complex agar biopolymer films with additives. While most formulations behave with a range of ultimate tensile strength between 0–15 MPa, there are some formulations that achieve between 20–40 MPa, similar to formulations with nanocellulose additives (e.g., numbers 23, 24, and 25).

As such, we find that by systematically varying the quantity of agar and glycerin, tensile properties can be achieved for simple agar biopolymer films that are comparable to those with more complex formulations. This underscores the importance of exploring the range of performance that is possible for even basic formulations.

Understanding the range of possible performance for a simple formulation can help in navigating trade-offs when trying to optimize a material along certain property dimensions by adding new ingredients. For example, the inclusion of cellulose nanocrystals extracted from onion peel improves resistance to breakage but decreases deformation capacity considerably [32]. Given such tradeoffs, starting with a basic formulation that achieves higher performance on a particular dimension before adding an ingredient could be useful. This highlights the utility of the data presented as a reference dataset that can be used for further material optimization. Notably, the materials reviewed may have other properties, such as water, gas barrier, and antimicrobial properties, that are being optimized for. Given this, as mechanical performance is fundamental to the viability of a biopolymer film for packaging, this study can provide a first screen for promising candidates that can then undergo further testing.

### 3.4. Comparison to Commercial Polymer Systems

Figure 4 illustrates how the data on agar biopolymer films compare to a range of commercially available polymers in terms of Young’s modulus, ultimate tensile strength, and elongation at break. Experimental data are demarcated in black and labelled Agar, with error bars signifying the range of values around property measurements. Commercial polymers used in plastic films were selected to make the comparison.

Figure 4a shows the Young’s modulus and elongation at break, and one can see that while the outer range of this study’s ‘Agar’ formulations is comparable to low density polyethylene (LDPE), they have low values of Young’s modulus compared to commercial petrochemical plastic film-forming materials.

The exception is hydroxyethyl cellulose (HEC) which is used as a binder, film former, and thickener for such applications as paints, textiles, cement, and architectural coatings. While the values of Young’s modulus found in this study are below those of commercial petroleum-based polymer films, they nonetheless are in a comparative range to other starch-based edible films and coatings, found to usefully extend the shelf life of fruits and vegetables [33,34,35,36,37,38]. Thus, such values of Young’s modulus are aligned with an important segment of single-use packaging applications.

Figure 4b showing ultimate tensile strength and elongation at break demonstrates a closer comparison with commercial polymers. Acrylonitrile butadiene styrene (ABS), thermoplastic starch (TPS), and polypropylene (PP) are all commercial plastics that overlap with the mechanical properties of the experimental dataset of agar biopolymer films.

Figure 4 enables the identification of potential applications based on comparability with the properties of commercial materials. For instance, PP is used for thin flexible packaging including shrink-film overwrap and food packaging. ABS is used in durable packaging for toys and cosmetics and a range of household goods. TPS, a bio-based commercial material, is used in packaging such as grocery bags. Select formulations also have similar ultimate tensile strength and elongation at break to hydroxypropyl cellulose (HPC), used as a binder and in the formation of films in pharmaceutical applications.

However, some overlapping polymer types may be primarily selected for properties other than their tensile properties. For instance, ABS is resistant to water and can be processed using a number of standard industrial processes such as injection molding [39]. To obtain a more complete picture of potential applications for the agar biopolymer films presented, further testing is needed to ascertain thermal, water, and gas barrier, and rheological properties among others. Thus, one should interpret these results as a promising first step in the identification of potential product applications based on tensile properties alone.

### 3.5. Neural Network Predictions

Mapping the parameter space of mechanical properties for the set of agar/glycerin/water formulations developed in this study can be used to suggest novel formulations with target mechanical properties. In an effort to explore this proposition, a non-linear regression model is implemented with a neural network to learn the relationship between three measured mechanical properties (Young’s modulus, elongation at break and ultimate tensile strength) and the composition of agar, glycerin, and water across the range of fabricated biopolymer films. No information regarding the fabrication or testing parameters (e.g., temperature, humidity) was provided to the neural network.

Figure 5 shows how ingredients concentrations are distributed across the range of formulations. Fewer data records are found at the tails of the histograms, which is shown to give lower predictive accuracies in the regression analysis below. In addition, a range of parameters could not be obtained on the basis that some samples were not amenable to tensile measurements, e.g., for glycerin from 30 mL to 50 mL or low water concentration (<200 mL).

Figure 6 illustrates the results of the model described in Section 2.6. Each data point represents a learning vector. Learning results are better, i.e., have low error values, for the points around the diagonal. Note that the learning errors of the points on the diagonal are zero. For example, Figure 6a shows larger prediction errors for higher values of Young’s modulus (*E* > 4, shaded area). This is because these measurements correspond to the lower and higher end of the agar and water concentration ranges studied, so few biopolymer film samples could be produced and tested. Higher Young’s modulus corresponds to formulations with relatively low water content in comparison with agar, which results in a stronger biopolymer film.

Considering the results in Figure 6, it is possible to conclude that the number of available measurements is large enough for the neural network architecture to learn the correlations between the inputs (biopolymer film composition) and outputs (mechanical properties) encoded in the dataset. The latter is attributed to the relatively high level of redundancy in the measurements, i.e., each film formulation is tested several times to build measurement statistics. Although the errors suggest that no additional input–output correlations can be inferred with the implemented network, the magnitude of the testing and validation error metrics cannot be considered optimal in the grey shaded regions of the parameter space.

## 4. Discussion and Conclusions

The goals of this study were to explore the mechanical performance range of simple agar-glycerin biopolymer films and compare them to other agar-based formulations and commercial polymer systems. In doing so, we aimed to understand the limits of mechanical performance that could be achieved with a simple process and set of ingredients and develop a reference dataset that could be used for further material optimization. A further aim was to explore the use of machine learning for predictive modelling, helping to define the utility of such methods for biopolymer film fabrication.

Aligned with the main goals, this study provides a broad set of agar biopolymer film formulations that explore a range of mechanical performance using the basic ingredients of agar, glycerin, and water. Moreover, at different ranges of ingredients concentrations, we find various optimal ranges in terms of the mechanical properties of interest. This illustrates the importance of exploring a wide range of ingredients concentrations when assessing how certain ingredients influence a material property of interest.

Of importance is the finding that measures of ultimate tensile strength and elongation at break can be achieved for simple formulations that are comparable or outperform more complex agar biopolymer films with a variety of additives and more elaborate processing steps. The importance of this finding is two-fold. First, the design space for simple agar biopolymer films is expanded, adding further information that can guide the identification of potential applications. Second, the range of mechanical performance achieved illustrates the utility of the study as a reference dataset, where select formulations that maximize tensile strength or elongation at break may be used as a starting point for further optimization efforts.

Furthermore, select formulations of simple agar biopolymer films exhibit values of ultimate tensile strength and elongation at break that are comparable to commercial petroleum-based plastic films, such as polypropylene (PP), thermoplastic starch (TPS), and acrylonitrile butadiene styrene (ABS). While Young’s modulus values are below most commercial polymer film values, they are nonetheless similar to other starch-based edible films and coatings studied in the literature [33,34,35,36,37,38], indicating promise for this type of application. Moreover, contrasting our results with agar-based films in the literature demonstrate the potential to improve Young’s modulus with additives and additional process steps. Given the simplicity of ingredients and process technique used in this study, these results hold promise for the future of seaweed bioplastics, where select formulations can compare favorably on key dimensions of material performance necessary for packaging applications.

While the formulations in this study may achieve good strength and ductility, the comparison with more complex agar-based formulations illustrates how a range of additives may improve elasticity. Moreover, these additives may help maximize other properties of interest, such as water or gas permeability or antimicrobial properties. The complexity of designing materials in this high dimensional space emphasizes the importance of data-driven approaches to materials design and optimization. Toward this end, the results from our preliminary neural network are promising; while a larger dataset is needed for validation, the results clearly support the further application of such methods to biopolymer-based formulations and the utility of predictive models.

### Future Research

Further research is needed on how agar biopolymer films and seaweed biopolymer films are more broadly compared with petroleum plastics over a range of material properties. In addition to mechanical properties, others such as water vapor permeability and antimicrobial properties are necessary to assess utility for single-use packaging applications, particularly in the food sector. Water permeation through a polymeric film implies three mechanisms: water absorption at the polymeric surface, diffusion throughout the polymeric film, and water desorption from the surface [40]. Despite being out of the scope of this work, water resistance is an essential aspect to be considered for biopolymer-based packaging materials [41]. Specifically, the use of additives to improve water resistance is an important avenue of study. For instance, [42] found that water resistance was optimal with native agar films with 50 or 75% of locust bean gum content, and [43] demonstrated that water resistance increases by adding lipids to agar-based biopolymer films. Solvent resistance, while also beyond the scope of this work, is also an important area of research [44]. While studies have indicated that additives such as polymer-clay nanocomposites can improve solvent resistance in some biopolymer systems [45], further work is needed to investigate in the context of agar biopolymer films and seaweed biopolymer materials more broadly.

To assess pathways for commercialization, it is also important to understand how processing parameters affect material properties, and whether seaweed biopolymers can be processed with a range of industrial methods such as compression and injection molding, casting, and blown-film extrusion.

Further research to expand the amount of data on agar biopolymer films would also be advantageous. For instance, characterizing the mid-points of the range of formulations in this dataset would improve the resolution of the response surfaces generated. Calculating the gel content or insoluble fraction is also important, where the degree at which cross linking occurs between polymers can be determined. The extent at which polymers are cross-linked is a key parameter influencing material properties of interest, including mechanical and barrier properties. Furthermore, research is needed into how seaweed species heterogeneity and biopolymer extraction methods affect material properties. This is particularly critical when studying materials such as agar-based biopolymer films where agar can be derived from a variety of species of red seaweed, using a variety of extraction methods. There is a need to generate larger datasets that combine this information with compositional variation, additives, and process parameters. Such data could be used to develop robust machine learning models that provide guidance on specific formulations that meet performance targets. Such a data-driven approach could help expediate the development of biodegradable seaweed-based plastics as potential replacements for petrochemical plastics.

## Figures and Tables

**Figure 1 materials-15-03954-f001:**
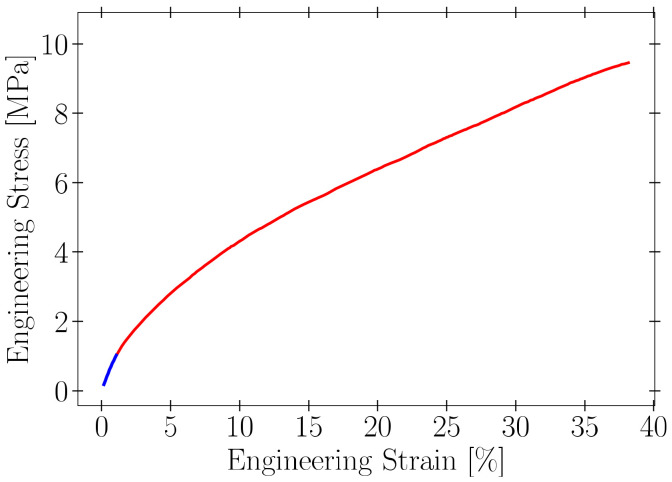
Average engineering stress as a function of engineering strain for a concentration of 2% agar and 1.25% glycerin. The blue line represents the linear portion or elastic zone, and the grey shaded region corresponds to the experimental error.

**Figure 2 materials-15-03954-f002:**
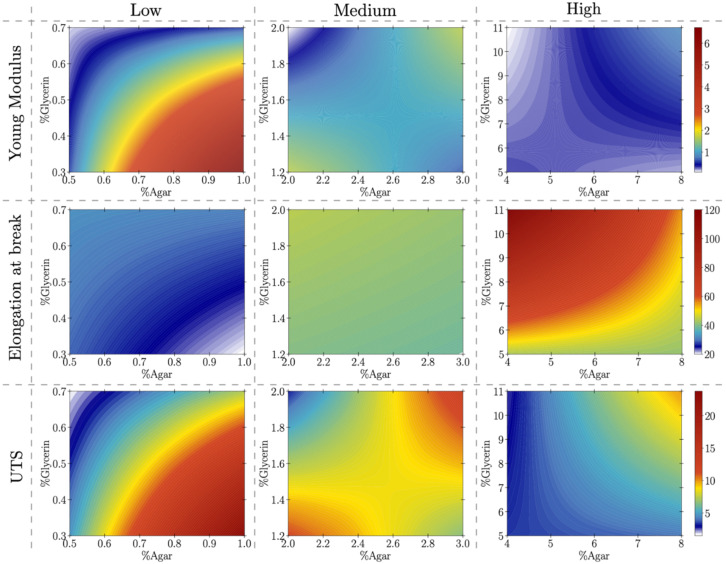
Linear portion of mechanical properties as a function of different concentrations of agar–glycerin percentages. The mechanical properties are the Young’s modulus (**left column**), elongation at break (**mid column**) and ultimate tensile strength (**right column**) for low (**top row**), medium (**mid row**) and high (**bottom row**) agar and glycerin concentrations. The regression coefficients for mechanical properties [Appendix A] are given in Appendix A.

**Figure 3 materials-15-03954-f003:**
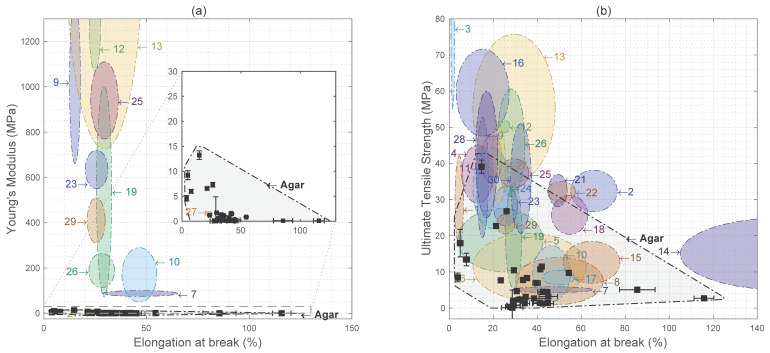
Comparison of results with agar-based films, illustrating (**a**) Young’s modulus versus elongation at break, and (**b**) ultimate tensile strength versus elongation at break.

**Figure 4 materials-15-03954-f004:**
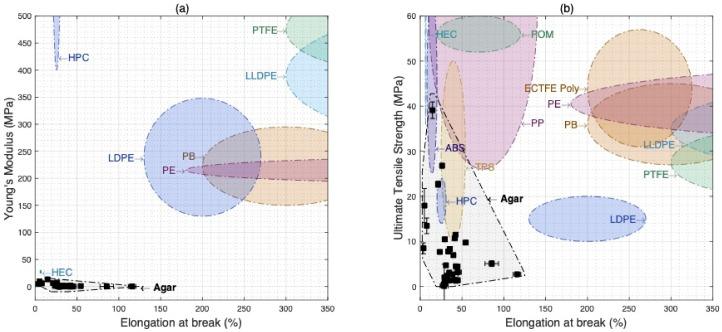
Material selection charts comparing agar biopolymer films to commercial polymer systems, illustrating (**a**) Young’s modulus versus elongation at break, and (**b**) ultimate tensile strength versus elongation at break.

**Figure 5 materials-15-03954-f005:**
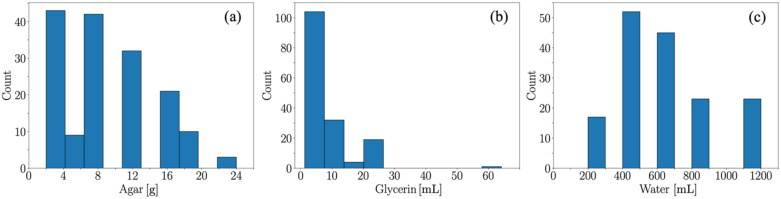
Distributions of (**a**) agar in grams, (**b**) glycerin in mL, and (**c**) water in mL used in the DoE to generate the biopolymer films.

**Figure 6 materials-15-03954-f006:**
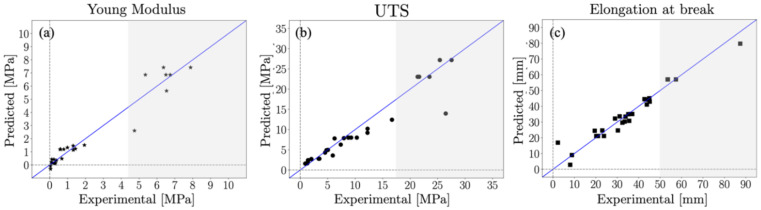
Learning results of (**a**) Young’s modulus (σ=0.89) (**a**,**b**) ultimate tensile strength (σ=2.78) and (**c**) elongation at break (σ=4.09). The diagonal blue line represents the position with zero error and the gray shaded area corresponds to regions in the film composition space with the smallest number of fabricated samples.

**Table 1 materials-15-03954-t001:** Input data for Design of Experiments (DoE).

Range	Value 1	Value 2	Agar (%)	Gly (%)
Low range	-	-	0.50	0.31
	+	-	1.00	0.31
	-	+	0.50	0.63
	+	+	1.00	0.63
Medium range	-	-	2.00	1.25
	+	-	3.00	1.25
	-	+	2.00	1.88
	+	+	3.00	1.88
High range	-	-	4.00	5.35
	+	-	8.00	5.35
	-	+	4.00	10.70
	+	+	8.00	10.70

**Table 2 materials-15-03954-t002:** Formulations and properties of agar biopolymer films (±standard deviation).

Agar (%)	Gly (%)	Young’s Modulus (MPa)	Elongation at Break [%]	Ultimate Tensile Strength (MPa)	No. Specimens
0.5	0.31	0.57 ± 0.08	30.71 ± 0.74	4.70 ± 0.28	5
0.5	0.63	0.23 ± 0.07	31.92 ± 3.40	2.37 ± 0.35	6
1	0.31	6.56 ± 0.64	21.29 ± 1.96	22.69 ± 1.47	6
1	0.63	1.35 ± 0.30	32.96 ± 2.91	8.76 ± 1.43	17
2	1.25	1.54 ± 0.12	41.4 ± 1.96	10.74 ± 0.79	6
2	1.88	0.33 ± 0.05	44.59 ± 0.84	4.45 ± 0.10	5
3	1.25	0.74 ± 0.35	39.72 ± 3.85	6.69 ± 0.68	5
3	1.88	1.50 ± 0.16	42.07 ± 1.88	11.48 ± 0.45	6
4	5.35	0.18 ± 0.24	41.78 ± 11.81	2.94 ± 0.88	13
4	10.7	0.08 ± 0.03	115.80 ± 12.75	2.71 ± 0.45	2
8	5.35	0.23 ± 0.00	42.51 ± 5.10	4.50 ± 0.38	4
8	10.7	0.83 ± 0.25	54.36 ± 1.75	9.74 ± 0.69	5

**Table 3 materials-15-03954-t003:** Range of mechanical properties of agar biopolymer films (±standard deviation).

Property Range	Young’s Modulus (MPa)	Elongation at Break (%)	Ultimate Tensile Strength (MPa)	Agar (%)	Gly (%)
Elongation at break (Min)	4.59 ± 1.21	3.92 ± 1.33	8.48 ± 2.69	2.00	0.31
Elongation at break (Max)	0.08 ± 0.03	115.80 ± 12.75	2.71 ± 0.45	4.00	10.70
Ultimate tensile strength (Min)	0.03 ± 0.00	28.66 ± 0.00	0.11 ± 0.00	1.00	10.70
Ultimate tensile strength (Max)	13.24 ± 1.92	14.68 ± 2.72	39.11 ± 4.02	1.00	0.28
Young’s modulus (Min)	0.03 ± 0.00	28.66 ± 0.00	0.11 ± 0.00	1.00	10.70
Young’s modulus (Max)	13.24 ± 1.92	14.68 ± 2.72	39.11 ± 4.02	1.00	0.28

## Data Availability

Data supporting reported results can be found at https://data.mendeley.com/datasets/nyz4y58jbt/1 (19 May 2022).

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
