# Peer review of "Agar Biopolymer Films for Biodegradable Packaging: A Reference Dataset for Exploring the Limits of Mechanical Performance"

_materials, 2022, doi:10.3390/ma15113954_

Round 1

Reviewer 1 Report

This work disclosed the mechanical properties of Agar-based films. Plastics have been widely used, but they are difficult to be degraded in environment and depend on fossil resources. In this work, authors produced Agar-based films and their mechanical performance was investigated. Although it has some significance especially from the sustainable development of packaging films, the original version of this manuscript cannot be published in Materials, revisions are necessary. Some comments and suggestions are as follows.

  1. The axis title in Figure 1 is not clear, what’s the meaning of them?
  2. Table 1 227 shows the relationships between measured mechanical properties by noting the minimum 228 and maximum values for each variable tested (elongation at break, UTS and Young's mod-229 ulus) across the range of formulations fabricated. I cannot see the formulations in the manuscript. To be clearly understood, the formulations and their related properties should be provided for each sample in a table.
  3. For the mechanical property test, the number of the repeated specimens for each sample should be provided in the “5. Characterization of mechanical properties”, and the standard deviations of the mechanical properties should be provided for the value of mechanical properties.
  1. In Figure 3, the results in this work is not clearly marked out.
  1. From the results in this work, I don’t know the advance of this work. Especially from Figure 4, of Agar-based films showed much lower properties than other plastic films. Authors should clearly point out the novelty or advance of this work in the manuscript.
  1. Authors mentioned the gelation of samples. Did chemical reaction occur in the gelation? What’s the gel content of the samples?
  1. How about the solvent or water resistance of the Agar-based films? authors should be investigated and discussed in manuscript.

Reviewer 2 Report

Although the article is well organised into different section but still leaves space for specific doubts, which needs to be clarified before considering for publication:

  1. The use of Circular Economy in the title is a catch, but how do the authors justify its inclusion with available recycling facilities?
  2. The minimum and maximum concentrations of each component are missing. Please give the input data chart prepared by/ for DoE.
  3. The study should include a middle level, apart from high and low.

Reviewer 3 Report

Comments on the Manuscript “Agar Biopolymer Films for a Circular Economy: A Reference Dataset for Exploring the Limits of Mechanical Performance” referenced by materials-1732196

This manuscript reports on a factorial experimental design to analyse how agar-based biopolymer films perform across a range of ingredients concentrations in terms of their strength, elasticity, and ductility. This manuscript should be deeply revised before acceptance for publication.

  1. “Agar Biopolymer Films for a Circular Economy” should be avoided in the title as far as this paper is mainly focus on a comparative evaluation of the mechanical properties of various agar-based biopolymer films and commercial polymer systems along with a neural network prediction.
  2. The abstract need to be more focused on the data presented in this manuscript. The novelty of this work should be better emphasized. In the current form, the abstract contains too many sentences related to packaging applications without a clear connection with the data presented in this manuscript. In addition, the authors should clarify if this manuscript is a review or a full length article.
  3. The term “machine learning” should be better described and associated with the data presented in the manuscript.
  4. The whole manuscript should be thoroughly checked. There are a lot of English language mistakes. Please avoid repetition. It is really hard to understand some parts of this manuscript due to English language mistakes.
  5. It is not clear which kind of data contains this manuscript. The authors should clarify if they discuss the already published data in the field or present new ones.
  6. The introduction should be deeply revised because it’s not related to this study. For instance, lines 72-74: “Agar has been blended with other biopolymers such as 72 starch [26–33], chitosan [34–36], microcrystalline cellulose [37], [38], lignin [39], locust 73 bean gum [40], carrageenan [41], gelatin [32], [42–46], soy protein isolate [47–50], and fish 74 protein hydrolysate [51].” Why the agar was blended with other biopolymers? Connection between the introduction paragraphs and sentences should be included. After each paragraph the authors should draw some advantages or disadvantages of the materials presented.
  7. The conclusions in their current state are a summary of the results. A conclusion should, however, summarize the results and set them into perspective to the objectives formulated in the introduction. Moreover it has to give an outlook on the importance of the findings. So, the authors should reword/amend the conclusions accordingly. Maybe a perspective section should be included.

Finally, I consider that this manuscript presents interesting and valuable data for the scientific audience, and it would be a pity if unclear sentences will diminish its value.

Round 2

Reviewer 1 Report

Authors have revised the manuscript according to the comments, thus, i think it can be published in Materials.

Reviewer 2 Report

The authors have answered all the queries.